# Change in Renal Glomerular Collagens and Glomerular Filtration Barrier-Related Proteins in a Dextran Sulfate Sodium-Induced Colitis Mouse Model

**DOI:** 10.3390/ijms20061458

**Published:** 2019-03-22

**Authors:** Chia-Jung Chang, Pi-Chao Wang, Tzou-Chi Huang, Akiyoshi Taniguchi

**Affiliations:** 1Cellular Functional Nanobiomaterials Group, Research Center for Functional Materials, National Institute for Materials Science, 1-1 Namiki, Tsukuba, Ibaraki 305-0044, Japan; CHANG.Chiajung@nims.go.jp; 2Graduate School of Advanced Science and Engineering, Waseda University, 3-4-1 Okubo, Shinjuku-ku, Tokyo 169-8555, Japan; 3Graduate School of Life and Environmental Sciences, University of Tsukuba, 1-1-1 Tennoudai, Tsukuba, Ibaraki 305-8572, Japan; wangpicao@gmail.com; 4Department of Biological Science and Technology, National Pingtung University of Science and Technology, Neipu, Pingtung 912-01, Taiwan; tchuang@mail.npust.edu.tw

**Keywords:** inflammatory bowel disease (IBD), DSS-colitis, glomerular filtration barrier (GFB), type IV collagen, type I collagen, type V collagen

## Abstract

Renal disease is not rare among patients with inflammatory bowel disease (IBD) and is gaining interest as a target of research. However, related changes in glomerular structural have rarely been investigated. This study was aimed at clarifying the changes in collagens and glomerular filtration barrier (GFB)-related proteins of glomeruli in a dextran sulfate sodium (DSS)-induced colitis mouse model. Acute colitis was induced by administering 3.5% DSS in Slc:ICR strain mice for eight days. Histological changes to glomeruli were examined by periodic acid-Schiff (PAS) and Masson’s trichrome staining. Expressions of glomerular collagens and GFB-related proteins were analyzed by immunofluorescent staining and Western blot analysis. DSS-colitis mice showed an elevated disease activity index (DAI), colon shortening, massive cellular infiltration and colon damage, confirming that DSS-colitis mice can be used as an IBD animal model. DSS-colitis mice showed increased glycoprotein and collagen deposition in glomeruli. Interestingly, we observed significant changes in glomerular collagens, including a decrease in type IV collagen, and an increment in type I and type V collagens. Moreover, declined GFB-related proteins expressions were detected, including synaptopodin, podocalyxin, nephrin and VE-cadherin. These results suggest that renal disease in DSS-colitis mice might be associated with changes in glomerular collagens and GFB-related proteins. These findings are important for further elucidation of the clinical pathological mechanisms underlying IBD-associated renal disease.

## 1. Introduction

Inflammatory bowel disease (IBD) is a chronic, remitting and relapsing inflammatory disease of the gastrointestinal tract characterized by inflammation and mucosal tissue damage and is associated with significant morbidity. Ulcerative colitis and Crohn’s disease are the two most common forms of IBD. Ulcerative colitis and Crohn’s disease differ from each other in physiology, but show similar symptoms such as severe diarrhea, rectal bleeding, abdominal pain, fever, and weight loss [1,2]. Clinical and epidemiological evidence suggests that IBD is a systemic disorder that can affect almost every organ [2,3]. Renal manifestations and complications in patients with IBD are not rare, and numerous clinical studies have reported that 4–23% of IBD patients experience renal disease such as tubulointerstitial nephritis, nephrolithiasis, and glomerulonephritis [4,5,6,7,8,9,10,11], which eventually induce renal disease. The appropriate experimental animal model of IBD-associated renal disease thus has clinical importance for related studies, including pathological mechanisms, prevention and treatment strategies for IBD.

Dextran sulfate sodium (DSS) is a water-soluble sulfated polysaccharide. Oral administration of DSS to trigger acute colitis has been widely used in experimental animal models for preclinical studies of IBD, because the pathophysiology resembles human Ulcerative colitis [12,13,14]. Recent studies have reported that mice with colitis induced by DSS show renal tubular injury that might be associated with increased neutrophil infiltration and expressions of cytokines and chemokines in both intestines and kidneys [15,16]. However, these studies of DSS-related renal injury have not mentioned structural changes to the glomeruli. On the other hand, the renal glomerulus is included in the nephron with the tubule, and tubular necrosis has been reported to potentially lead to declines in glomerular function [17,18], suggesting that glomerular damage might be accompanied by tubular injury. The glomerulus contains a highly specialized filtration barrier structure that is essential for maintaining normal plasma ultrafiltration, and loss of glomerular filtration function can lead to poor blood filtration, resulting in renal disease [19,20]. The electively permeable glomerular filtration barrier (GFB) is a three-layered structure that separates the capillaries and Bowman’s space, and comprises the interdigitating foot processes of the podocytes, the intervening glomerular basement membrane (GBM), and the fenestrated endothelium [21,22]. The GBM is constituted of specialized extracellular matrix (ECM) components, namely type IV collagen, laminin and other proteoglycans, which are essential for providing a complete structural scaffold to the glomeruli, and important for establishing and maintaining the integrity of the GFB [23,24]. Disrupted GBM has been demonstrated to lead to filtration barrier damage and eventual glomerular disease [24,25]. However, whether GFB-related proteins changes are involved in IBD associated renal disease has not yet been clarified. Herein, we hypothesized that DSS induces renal structural changes, particularly to the glomerular structure.

In this study, we investigated glomerular structural changes focusing on specific types of glomerular collagens and GFB-related proteins after DSS administration, to demonstrate the coexistence of glomerular structural changes and IBD in a DSS-induced colitis mouse model. This study should help establish an experimental animal model for further elucidation of the clinical pathological mechanisms of IBD-associated renal disease.

## 2. Results

### 2.1. Progression of DSS-Colitis

After DSS administration (Figure 1), significant body weight loss was observed on Days 4–8 as compared to those of controls (water only) (*p* < 0.05) (Figure 2A). DAI scores showed elevating values after three days of DSS administration, reaching a peak on Day 8 (Figure 2B). Colon shortening, a marker of the severity of colorectal inflammation [14], was significantly greater in DSS-colitis mice as compared to the control group by Day 8 (Figure 2C). Histological observation of the colon was subsequently performed using HE staining, which showed a normal morphology of crypts, abundant goblet cells, muscular layer, submucosa and mucosa in the control mice. However, DSS-colitis mice revealed severe epithelial damage with mucosa thickening, massive cellular infiltration into the lamina propria and colon mucosa, crypt distortion, goblet cells loss, and complete destruction of the architecture (Figure 2D). These histological changes indicated severe inflammatory colitis.

### 2.2. Renal Morphology and Histological Changes in DSS-Colitis Mice

Human and mouse studies have indicated the involvement of non-intestinal organs in IBD [4,5,6,7,8,9]. We therefore investigated renal changes in DSS-colitis mice. Kidney size and weight of DSS-colitis mice were decreased in mice after eight days of treatment as compared to those in controls (Figure 3A,B). To detect structural histological damage, glomerular morphology was examined in tissue sections by PAS and Masson’s trichrome staining. Deep pink color (PAS-positive matrix) representing deposition of matrix glycoprotein was apparent at the GBM and mesangium, confirming increased matrix in the glomeruli of DSS-colitis mice (Figure 3C). Masson’s trichrome staining was performed to detect the collagen deposition and fibrosis associated with renal disease, and blue to blue-violet staining indicated the presence of collagen in tissues. The result showed lightly stained collagen in the GBM and tubular basement membrane. On the other hand, significant collagen deposition among the glomerular capillaries and surrounding the Bowman’s capsules were observed in DSS-colitis mice after eight days of DSS administration (Figure 3D) compared to control mice.

### 2.3. Collagen Changes in Glomeruli

Glomerular collagens including type IV collagen (a typical collagen of the basement membrane matrix), type I collagen (an interstitial matrix collagen) and type V collagen (an atypical collagen that only appears at kidney development and in kidney diseases such as collagenofibrotic glomerulopathy) [26,27] were investigated by immunofluorescent microscopy and Western blot analysis. The results showed decreased type IV collagen in GBM and Bowman’s capsules (Figure 4A-a’) in DSS-colitis mice as compared to controls (Figure 4A-a). In contrast to type IV collagen, type I and V collagens increased in the glomerular and renal interstitium (Figure 4A-b’,-c’) of DSS-colitis mice as compared to those in controls (Figure 4A-b,-c). Consistent with immunofluorescent microscopy results, Western blotting analysis also showed declining expressions of type IV collagen (Figure 4B-a), and increasing expressions of type I and V collagens (Figure 4B-b,-c) in the renal cortex of DSS-administered mice, indicating the influence of DSS on changes in glomerular collagens. The above results are illustrated in Figure 4C.

### 2.4. Changes in GFB-Related Proteins

The GFB comprises glomerular endothelial cells, the GBM and podocytes [22,23]. Podocytes and glomerular endothelial cells are located on opposite side of the GBM. Immunofluorescent investigation of podocyte-associated proteins in glomeruli (including synaptopodin, podocalyxin, and nephrin) were performed to detect changes in podocytes. The results showed that synaptopodin, podocalyxin, and nephrin were significantly expressed at capillary tufts of normal glomeruli in the control mice (Figure 5A-a, -b, -c), but declined in DSS-colitis mice after DSS administration (Figure 5A-a’, -b’, -c’). Similarly, the vascular-specific junctional molecule VE-cadherin in endothelial cells on the GBM, which is associated with the regulation of vascular permeability and glomerular filtration, showed lower immunofluorescence and discontinuous expression in DSS-colitis mice (Figure 5A-d’) as compared to that in controls (Figure 5A-d).

Western blot analysis of these four proteins in glomeruli showed declines in all proteins (Figure 5B), consistent with the results from immunofluorescence (Figure 5A).

These findings confirmed that DSS administration caused podocyte damage (reductions in synaptopodin, podocalyxin and nephrin) and changes to endothelial adherens junctions in glomerular endothelium (reductions in VE-cadherin). These results are shown in Figure 5C.

## 3. Discussion

Renal manifestations and complications in patients with IBD have been reported in several clinical and experimental studies from recent years [4,5,6,7,8,9,10,11,28]. However, the coexistence of renal disease and IBD, and the related glomerular structural changes in particular, have rarely been discussed. This study provided novel data showing not only the changes in renal glomerular collagen types in a DSS-induced colitis mouse model, but also revealed the interesting fact that proteins located on both the glomerular podocyte slit diaphragm and endothelial junction declined in expression in DSS-colitis mice, reflecting GFB damage.

This study adopted the DSS-induced colitis mouse model and confirmed symptoms of DSS-induced colitis such as body weight loss, diarrhea, gross bleeding and colon architecture destruction (Figure 1 and Figure 2), similar to IBD symptoms in humans [12,29]. Although some previous reports have mentioned that DSS might influence kidney function in mice after observing acute inflammatory responses associated with pro-inflammatory cytokine and chemokine expression in both intestines and kidneys, as well as renal tubular injury [15,16], the glomerular damage, especially the GFB damage in this DSS model still lack detailed evidence. In this study, we conducted detailed investigations into changes in the renal glomerular histology, GBM, and GFB-related protein expression, to illustrate renal damage in the DSS-induced colitis mouse model.

We noticed that kidney size and weight were decreased in DSS-colitis mice after DSS administration (Figure 3), since kidney size and weight are important indicators of renal pathology during disease development [30]. The decreased kidney size may correlate with body weight loss [30,31], and this phenomenon has been found in human patients with IBD [32]. Moreover, PAS staining revealed increasing deposition of glycoprotein matrix in GBM and mesangium, and Masson’s trichome staining revealed collagen deposition was markedly increased around the glomerulus and Bowman’s capsules in DSS-colitis mice. Such matrix and collagen depositions in glomeruli were also found in glomerular impairment, implicating excess ECM production as a factor in glomerular disease [27,33,34]. These observations suggest the DSS mice might show some glomerular abnormality.

To clarify the assumption that DSS induces renal structural change, we investigated the changes in specific types of collagens in the glomeruli. Our results showed that type IV collagen was decreased in the GBM, whereas type I and V collagens increased in the renal glomerular capillary loops and interstitium of DSS-colitis mice (Figure 4). Type IV collagen is well known to be the major component of the ECM in GBM, Bowman’s capsule and tubular basement membranes in normal kidneys [35]. Decreased expression of type IV collagen in GBM has been reported with the increased GBM degradation associated with kidney dysfunction [35,36,37,38]. On the other hand, type I and V collagens belong to interstitial ECM and excessive deposition is known to form scar tissue in the interstitial space during fibrosis [39,40]. In fact, type I collagen seldom appears in renal vessels and glomeruli under normal conditions, but is deposited in the early stage of renal fibrosis [27,39]. Type V collagen has been reported to spread widely in glomeruli during glomerulopathy, wound healing and kidney development [26,41,42]. Our result of a decrement in type IV collagen in GBM and increments in both type I and type V collagens in renal glomerular capillary loops and interstitium suggested that DSS administration could cause these collagens changes, which may lead to glomerular structure damage.

Renal disease has been reported in human IBD patients, but GFB-related protein changes have not been closely investigated. The present study investigated four proteins (synaptopodin, podocalyxin, nephrin and VE-cadherin) to clarify GFB damage, because these proteins have not been investigated in the kidneys of DSS-colitis mice, and the relevance of GBM damage in the DSS-induced colitis mouse model has not been reported yet. Our results showed declined expression of all four proteins in glomeruli after DSS administration (Figure 5). Podocytes locating on the GBM are known to serve as the final filtration barriers of glomeruli and contain the special proteins synaptopodin, podocalyxin and nephrin. These proteins have been suggested to represent important biomarkers of podocyte deficiency [43]. Synaptopodin is known to maintain podocyte foot processes and downregulation of the podocyte actin cytoskeleton has been observed in human and rodent glomerular diseases [43]. Decreased synaptopodin expression in podocytes reflects the foot processes are associated with a loss of cytoskeletal destruction [44]. Podocalyxin is a highly electronegative sialoglycoprotein located at the apical surface of podocyte foot processes and functions to maintain the negative charge of the glomerular filtration slit diaphragm and podocyte shape by linking to the actin cytoskeleton [45]. In vivo and in vitro studies have reported that decreased podocaylxin is associated with reduced adhesiveness of cells to the GBM [43,45]. Moreover, the loss of nephrin, a structural protein located between the podocyte foot processes, causes decreased podocyte integrity of slit diaphragms, resulting in eventual damage to the GFB [19,46]. Furthermore, glomerular endothelial cells serve as the first filtration barrier through their tight adhesion to the basement membrane, and a decrease in endothelial cells can therefore worsen renal failure [20,21]. VE-cadherin is an adherens junction protein between endothelial cells that maintains vascular integrity and decreased VE-cadherin expression has been observed in the glomerular endothelium of end-stage renal disease patients [47,48,49].

Taken together, our results imply that DSS administration could cause the glomerular collagen changes, including decreased type IV collagen as a supporting ECM of GBM structure and deposition of type I and type V collagens in renal interstitium. These collagen changes might lead to structural damage to podocytes such as a loss of polarity and detachment from the GBM, as well as loss of endothelial cell junctions, eventually causing renal disease. Loss of the podocyte cytoskeletal proteins synaptopodin and podocalyxin and the slit diaphragms protein nephrin, as well as the defective endothelial cells adherens junction protein (VE-cadherin) which may be associated with podocyte damage.

Based on our findings, the lack or insufficiency of these GFB-related proteins in DSS-colitis mice might cause glomerular structural damage, and consequently lead to damage not only to podocytes, but also to adherens junctions in the vasculature, which might result in GFB damage.

In conclusion, this study used the DSS-induced colitis mouse model, a very common experimental model of colitis, to clarify changes in glomerular collagens and GFB-related proteins after DSS administration. These findings on glomerular structural change in experimental mice with DSS-induced colitis should lead to novel uses of the animal model for further investigations into IBD-associated renal disease.

## 4. Materials and Methods

### 4.1. Laboratory Animals

Seven-week-old male Slc:ICR strain mice weighing 28–30 g (Japan SLC, Hamamatsu, Japan) were housed in the Central Animal House at the University of Tsukuba under climate-controlled conditions (room temperature, 22 ± 2 °C; 12-h light/dark cycle; relative humidity, 65%). Cages were cleaned and sterilized every week. All mice were given free access to food and water. Animal experiments were conducted in strict accordance with the recommendations of the Guide for the Care and Use of Laboratory Animals of the Science Council of Japan and Ministry of Education, Culture, Sports, Science and Technology of Japan. The protocol was approved by the Committee on the Ethics of Animal Experiments of the University of Tsukuba (Permit Number: 14-047; May 2016) based on the Institutional Animal Care and Use Committee (IACUC). All surgery was performed under sodium pentobarbital anesthesia, and all efforts were made to minimize suffering.

### 4.2. Induction of DSS-Colitis in Mice

As a widely used model of IBD, the DSS-induced colitis mouse model was used in this study to investigate changes to renal glomerular collagens and GFB-related proteins. This mouse model was established by inducing colitis with the administration of DSS in drinking water for eight days (Figure 1), following the previously described method [15]. Briefly, mice were administered 3.5% (*w*/*v*) DSS (MW 36,000–50,000 Da; MP Biomedicals, Solon, OH, USA) dissolved in drinking water for eight days. Six mice per group were used in each experiment, and were not allowed to access to any other source of water. Mouse weights were monitored daily to quantify the systemic consequences of colitis. Mice were fed with normal water in place of DSS water on Day 8 and euthanized on Day 9. Colon length and kidney weight were measured for each mouse at harvest, then fixed in 10% neutral buffered formalin (Wako, Osaka, Japan) or stored at −80 °C for further use. Mice receiving only distilled water were used as controls.

### 4.3. Disease Activity Index (DAI)

Weight loss, stool consistency, and rectal bleeding were recorded daily to assess the severity of DSS-colitis. DAI was determined based on the methods described previously [50]. Briefly, DAI was scored from 0 to 4 for each parameter, and then averaged for each group. Parameters were body weight loss (0 = no weight loss; 1 = 1–5% weight loss; 2 = 6–10% weight loss; 3 = 11–15% weight loss; and 4 = ≥ 15% weight loss), stool consistency (0 = normal stools; 2 = loose stools; 4 = diarrhea) and gross bleeding (0 = negative, 2 = positive occult blood in stools, 4 = rectal bleeding). DAI was calculated as the sum of the weight loss, stool consistency and gross bleeding scores.

### 4.4. Histological Investigation

To detect injury to the colon and renal tissues, prepared tissues were cut into 2-μm thick sections using a microtome (Thermo Fisher Scientific, Waltham, MA, USA) and stained with hematoxylin and eosin (HE) for histological investigations. To assess the GBM, periodic acid-Schiff (PAS) was used to estimate the glomerular deposition of matrix glycoprotein [51]. Briefly, 3-μm-thick sections were stained using a PAS staining kit (Merck, Darmstadt, Germany) and counterstained with hematoxylin according to the instructions from the manufacturer. To detect collagen fiber deposition [52], 3-μm sections were stained with a Masson’s trichrome staining kit (Muto Pure Chemicals, Tokyo, Japan) according to the manufacturer’s instruction. Microscopic images were acquired using a light microscope with a charge-coupled device camera (Olympus, Tokyo, Japan).

### 4.5. Immunofluorescence Staining and Confocal Imaging

Immunofluorescence staining was performed by following the method described previously [53]. Briefly, kidneys were embedded in optimal cutting temperature compound (Sakura Finetek, Tokyo, Japan), and frozen in liquid nitrogen. Sections of 5-μm thickness were cut by a cryostat (CM3050; Leica, Wetzlar, Germany); then incubated with primary antibodies against type I collagen (Acris Antibodies, Germany), type IV collagen, type V collagen, synaptopodin, VE-cadherin (Santa Cruz Biotechnology, Santa Cruz, CA, USA), nephrin, and podocalyxin (R&D Systems, Minneapolis, MN, USA), respectively, at 4 °C overnight, and followed by secondary antibodies conjugated to Alexa Fluor^®^ 488 or 568 (Invitrogen, Carlsbad, CA, USA); double-stained with rhodamine-conjugated phalloidin (Life Technologies, Gaithersburg, MD, USA) for F-actin; and finally submerged in fluoroshield mounting medium containing 4′,6-diamidino-2-phenylindole (DAPI) (Abcam, Cambridge, UK). Confocal imaging was performed according to the method described previously [26] with a confocal microscope (LSM700; Carl Zeiss, Jena, Germany). Alexa Fluor^®^ 488, and 568 signals were detected at laser excitation wavelengths of 488 nm and 543 nm, respectively.

### 4.6. Renal Glomerular Isolation

Kidney cortex tissue from each mouse was removed, glomeruli were isolated using a serial sieving method described previously [54], and then washed with PBS to remove small tubular fragments. The isolated glomeruli were re-suspended by PBS for further use or dissolved in lysis buffer for protein extraction.

### 4.7. Protein Isolation and Western Blot Analysis

Kidney cortices (~50 mg) or isolated glomeruli from kidney cortices of each mouse were homogenized in lysis buffer (50 mM Tris pH 7.4, 250 mM NaCl, 5 mM EDTA, 2 mM Na_3_VO_4_, 1 mM NaF, 20 mM Na_4_P_2_O_7_, 0.02% NaN_3_, 1% Triton X-100, 0.1% SDS and 1 mM PMSF) with 1% protease inhibitor cocktail (Sigma-Aldrich, St. Louis, MO, USA) by sonication (Qsonica, Newtown, CT, USA), followed by incubation on ice for 10 min and centrifugation at 10,000 rpm for 10 min. Supernatant was collected and protein concentration was quantitated using the Micro BCA Protein Assay kit (Thermo Fisher Scientific, Waltham, MA, USA) according to the instructions from the manufacturer. Thirty micrograms of total protein were loaded per lane and separated by 7.5% polyacrylamide gels, followed by transferring onto methanol-activated PVDF membrane (Millipore, Billerica, MA, USA). Protein-transferred membranes were incubated with primary antibodies against type I collagen, type IV collagen, type V collagen, synaptopodin, VE-cadherin, nephrin, podocalyxin and β-actin, respectively, at 4 °C overnight, followed by incubation with horseradish peroxidase-conjugated secondary antibody for 2 h at room temperature. Blots were visualized with chemiluminescence substrate for 1 min and detected using a luminescent image analyzer (LAS-4000 mini; Fujifilm, Tokyo, Japan). Band density was quantitated densitometrically with Image J software (National Institutes of Health, MD, USA) by calculating the average optical density in each band. Relative and normalized protein expressions were calculated using the ratio of each protein density to β-actin density.

### 4.8. Statistical Analysis

Statistical analyses were performed using Prism software (GraphPad, San Diego, CA, USA). All data are expressed as mean ± standard error of the mean (SEM) from 6 replicates (*n* = 6 mice per group) in at least 3 independent experiments. The significance of differences between groups were analyzed using Student’s *t*-test. A probability level of *p* < 0.05 was considered significant.

## Figures and Tables

**Figure 1 ijms-20-01458-f001:**
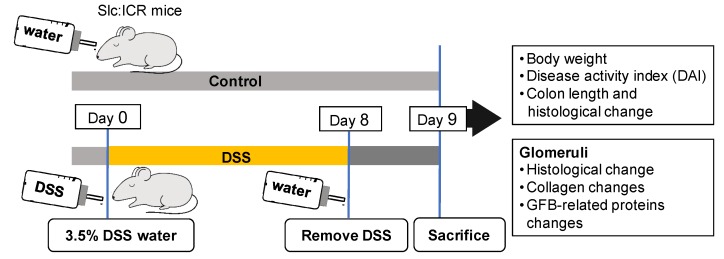
Investigating mouse glomerular structural changes associated with dextran sulfate sodium (DSS)-induced colitis. Slc:ICR mice were administered 3.5% DSS in drinking water for eight days, then allowed intake of filtered water on Day 8. Control mice were given filtered water. All mice were sacrificed on Day 9 and further assessments were performed. Abbreviation: GFB, glomerular filtration barrier.

**Figure 2 ijms-20-01458-f002:**
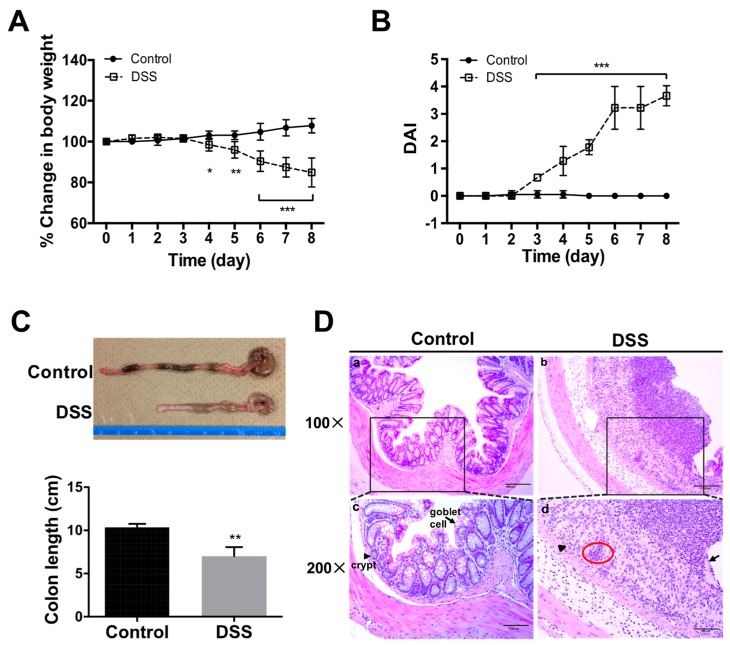
Macro- and microscopic changes to bowel in mice with DSS-induced colitis. Changes in body weight (**A**) and disease activity index (DAI) (**B**) were evaluated daily. Colon length was measured after sacrifice (**C**). Hematoxylin and eosin (HE) staining (**D**) showed distortion of crypts (arrowhead), loss of goblet cells (arrow), and infiltration of inflammatory cells (red circle) in colon sections from DSS-treated mice. All values are given as mean ± SEM (*n* = 6 mice); * *p* < 0.05, ** *p* < 0.01, and *** *p* < 0.001 vs. control. Scale bars: 200 μm (**a**,**b**); and 100 μm (**c**,**d**).

**Figure 3 ijms-20-01458-f003:**
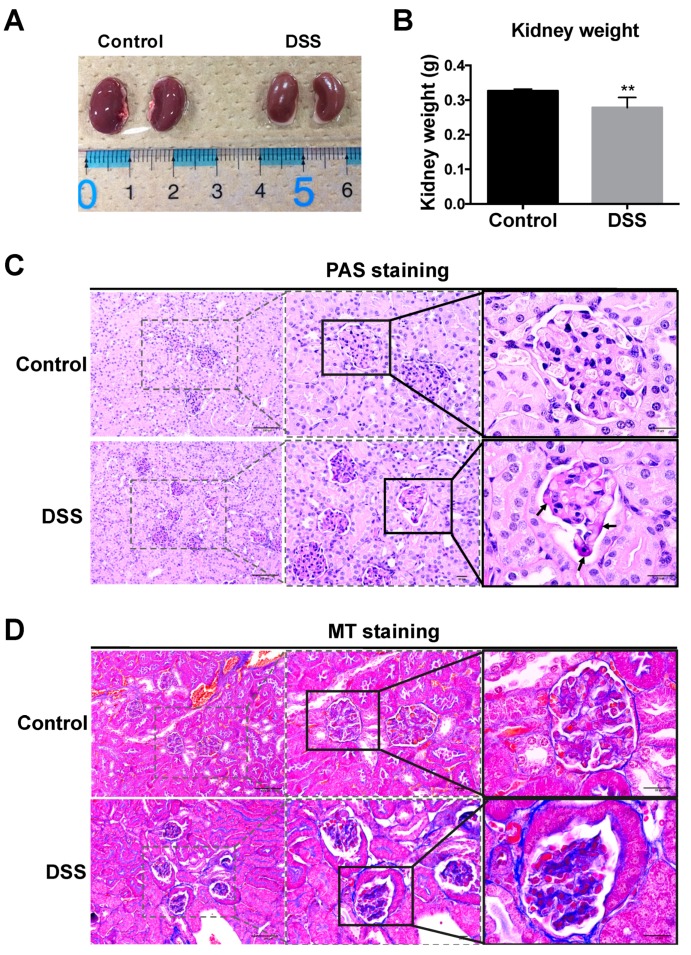
Macro- and microscopic changes to the kidney and glomeruli in mice after DSS administration. Mouse kidney appearance (**A**) and weight (**B**) were determined at harvest. Histological manifestations were determined by staining with periodic acid-Schiff (PAS) to assess the basement membrane of glomeruli (**C**), and Masson’s trichrome (MT) staining to assess collagen deposition (**D**), respectively. Compared to control mice, glomerular accumulation of PAS-positive matrix (arrow) was prominent in DSS-treated mice (**C**). Blue staining indicates the presence of collagen fibers in tissues (**D**). All values are given as mean ± SEM (*n* = 6 mice); ** *p* < 0.01 vs. control. Scale bars: 100 μm and 20 μm.

**Figure 4 ijms-20-01458-f004:**
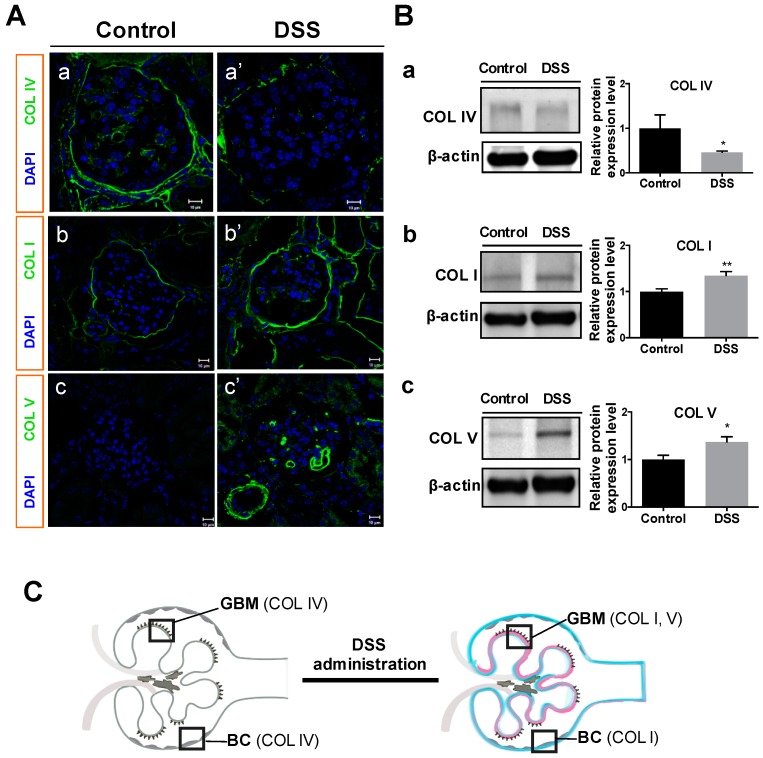
Changes in glomerular collagens in mice after DSS administration. Immunofluorescent microscopy (**A**) and Western blot analysis of protein expression (**B**) for type IV collagen (COL IV; A-a, A-a’; B-a; 160–190 kDa), type I collagen (COL I; A-b, A-b’; B-b; 150 kDa), and type V collagen (COL V; A-c, A-c’; B-c; 220 kDa) were conducted for control and DSS-colitis mice. Representative bands (**B**, left) and relative band intensity ratios were analyzed (**B**, right). (**C**) Illustration of glomerular collagens changes in this study. All values are means ± SEM (*n* = 6); * *p* < 0.05 and ** *p* < 0.01 vs. control. Scale bars = 10 μm. Abbreviations: GBM, glomerular basement membrane; BC, Bowman’s capsule.

**Figure 5 ijms-20-01458-f005:**
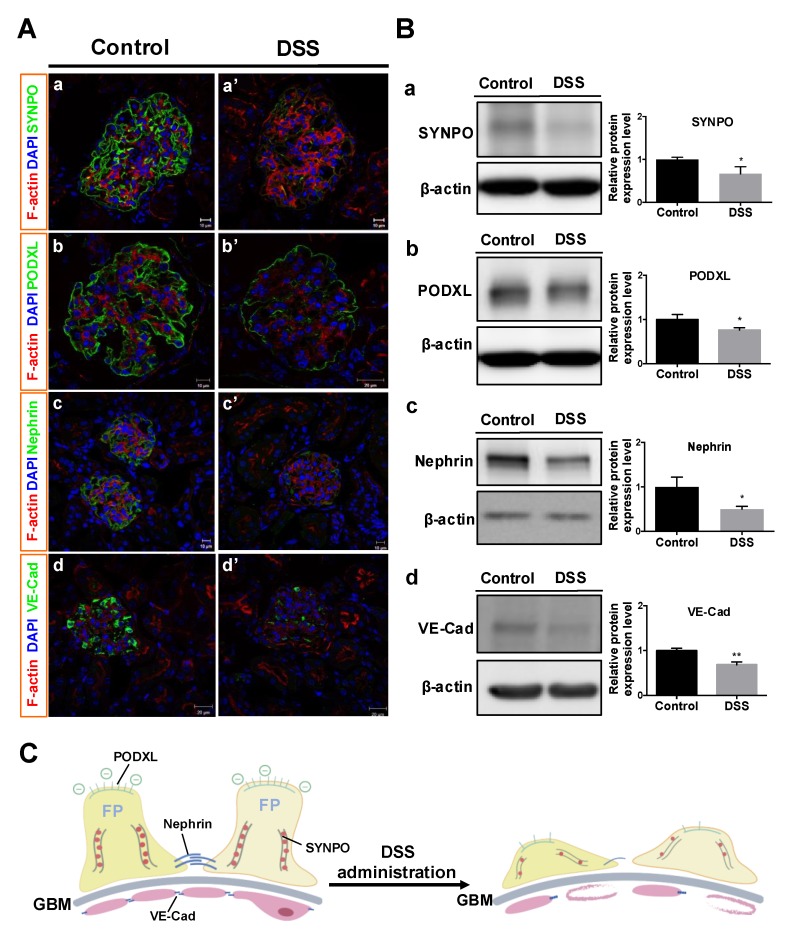
Changes in GFB-related proteins in mice after DSS administration. Immunofluorescent microscopy (**A**) and Western blot analysis of protein expression (**B**) against synaptopodin (A-a, A-a’; B-a; 100 kDa), podocalyxin (A-b, A-b’; B-b; 130 kDa), nephrin (A-c, A-c’; B-c; 185 kDa) and VE-cadherin (A-d, A-d’; B-d; 130 kDa) in glomeruli were conducted for control and DSS-colitis mice. (**B**) Representative bands (left), and relative band intensity ratios (right) were analyzed. (**C**) Illustration of GFB-related proteins changes in this study. All values are means ± SEM (*n* = 6), * *p* < 0.05 and ** *p* < 0.01 vs. control. Scale bars = 10 μm (A-a, A-a’, A-b, A-c, A-c’); 20 μm (A-b’, A-d, A-d’). Abbreviation: SYNPO, synaptopodin; PODXL, podocalyxin; VE-Cad, VE-cadherin; FP, foot processes.

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
