# Peer review of "Change in Renal Glomerular Collagens and Glomerular Filtration Barrier-Related Proteins in a Dextran Sulfate Sodium-Induced Colitis Mouse Model"

_ijms, 2019, doi:10.3390/ijms20061458_

Round 1

Reviewer 1 Report

The manuscript by Chang et al. describes alterations in glomeruli (increased glycoprotein and collagen deposition) in mice treated with 3.5% dextran sulfate sodium (DSS), which is an excellent IBD animal model. The authors also demonstrated significant alterations (decrease in type IV collagen and increment in type I and type V collagens) and GFB-related proteins (declined expression of synaptopodin, podocalyxin, nephrin and VE-cadherin) in glomeruli. The findings are of interesting and important for IBD patients. The experiments were carefully conducted and the manuscript is well-written. However, this reviewer would like to suggest that several papers can be added in the references. They include PMID: 30274631 and PMID: 24262508. Also, investigation on human samples and mechanistic studies are warranted.

Author Response

Thank you for your valuable advices. The two papers, “The histopathologic spectrum of kidney biopsies in patients with inflammatory bowel disease” and “Renal Manifestations of Inflammatory Bowel Disease”, have been included in the references cited in Introduction, Results and Discussion of the revised manuscript on lines 46, 90-91 and 172.

Reviewer 2 Report

The paper by Chang and colleagues concerning the connections between inflammatory bowel disease and glomerular barrier impairment  is well written. The methodology is robust and the results well explained and discussed.

very minor spelling mistake occurred on line 63: podocyte should be podocytes.

I would suggest the manuscript be accepted for publication as it stands. 

Author Response

Thank you for your helpful advice. We have replaced podocyte with podocytes on line 63.